# Evaluation of a 3D-Printed Reduction Guide for Minimally Invasive Plate Osteosynthesis of Short Oblique Radial Diaphyseal Fracture in Dogs: A Cadaveric Study

**DOI:** 10.3390/vetsci11040145

**Published:** 2024-03-22

**Authors:** Seungyeol Lee, Kangwoo Yi, Namsoo Kim, Suyoung Heo

**Affiliations:** Department of Surgery, College of Veterinary Medicine, Jeonbuk National University, Iksan-si 56896, Republic of Korea; dltmdduf92@jbnu.ac.kr (S.L.); dlrkddn430@jbnu.ac.kr (K.Y.); namsoo@jbnu.ac.kr (N.K.)

**Keywords:** 3D-printed reduction guide, minimally invasive plate osteosynthesis, radial fracture, dog, reduction

## Abstract

**Simple Summary:**

Radial and ulnar fractures often occur in the middle or distal third of the diaphysis. These fractures are common orthopedic injuries in dogs accounting for up to 17% of canine fractures. The limited soft tissue coverage and intraosseous circulation of the distal third of the diaphysis of the radius may contribute to a higher frequency of healing complications. Minimally invasive plate osteosynthesis (MIPO) techniques can help rapid bone healing. However, the application of MIPO is difficult in attaining and maintaining bone alignment. To compensate for these problems, this study designed a patient-specific 3D reduction guide to align radial diaphyseal fractures using the MIPO method. In this study, we found that the use of a patient-specific 3D reduction guide during MIPO of short oblique radial diaphyseal fractures in dogs is reliable for the alignment and apposition of fractures and reduces the surgical time.

**Abstract:**

This study aims to evaluate the clinical application of three-dimensional (3D)-printed custom reduction guides (3DRG) for minimally invasive plate osteosynthesis (MIPO) of short oblique radial diaphyseal fractures. Canine forelimb specimens (*n* = 24) were prepared and a diaphyseal short oblique fracture was simulated in the distal radius and ulna. Bone fragments were stabilized with the MIPO technique using a 3DRG (Group A), open reduction (Group B), or closed reduction with circular external skeletal fixation (ESF) (Group C). The diaphyseal short oblique fractures were created in each radius at one-third of the radial length from the distal radial articular surface. Surgical stabilization of the fractures was performed in each group. Pre and postoperative radiographic images were obtained to measure frontal angulation (FA), sagittal angulation (SA), frontal joint reference line angulation (fJRLA), sagittal joint reference line angulation (sJRLA), translational malalignment and fracture gap width. Surgical time was also measured. In the homogeneity test, differences in SA, sJRLA, craniocaudal translation and fracture gap before and after surgery had no significant difference among the three groups. On the other hand, differences in FA, fJRLA, mediolateral translation and surgical time before and after surgery had significant differences among the three groups. In the post hoc test, only surgical time showed a significant difference between the three groups, and group A showed the shortest surgical time. The use of 3DRG for MIPO of short oblique radial diaphyseal fractures in dogs is reliable for the alignment and apposition of fractures and reduces surgical time.

## 1. Introduction

Radius and ulna fractures often occur in the middle or distal third of the diaphysis and frequently involve both the radius and the ulna. These fractures are common injuries, representing approximately 8.5–17% of all fractures incurred by dogs [1,2]. The minimal soft tissue coverage and poor intraosseous vascularity of the distal third of the diaphysis of the radius may result in small to toy-breed dogs being predisposed to healing complications [1]. Postoperative complications of radial and ulnar fractures in these dogs include delayed union and non-union [1].

Surgical stabilization of radius and ulna fractures in dogs and cats most commonly uses the application of a bone plate and screws. Traditionally, bone plate and screw applications have used an open surgical approach and direct fracture reduction, often resulting in soft tissue trauma and devascularization of bone fragments at the fracture site [3]. In humans, but more recently in dogs, minimally invasive plating techniques have been developed to preserve the fracture hematoma and local blood supply [1,2,3,4,5]. Minimally invasive plate osteosynthesis (MIPO) involves the application of a plate through small incisions made remote to the fracture site, followed by the development of an epiperiosteal tunnel between these incisions. This technique can help by allowing a faster return to function by preserving more periosteal blood supply around the fracture site and minimizing damage to soft tissues [1,2,3,4,5].

MIPO is primarily applied to comminuted fractures, but it can also be applied to transverse, short oblique and spiral fractures [6]. The methods used in the closed reduction for MIPO include manual traction, hanging the limb, the use of intramedullary pins, application of a pre-contoured plate, distraction tables, temporary external fixators, fracture distractors and percutaneous placement of reduction forceps [6,7,8]. Despite these numerous methods, the clinical application of MIPO for the stabilization of diaphyseal fractures in veterinary surgery is not widely used. A major barrier to the widespread application of MIPO is the difficulty in attaining and maintaining bone alignment before internal fixation [8]. Another barrier is that MIPO does not allow for the direct observation of fracture fragments during surgery, making it necessary to use fluoroscopy to evaluate the precise reduction in fracture fragments. However, fluoroscopy significantly increases the amount of radiation to which the surgical team and the patient are exposed [5].

Three-dimensional (3D)-printed guides have been used in veterinary surgery for fracture repairs. One veterinary orthopedic cadaveric study reported a 3D-printed patient-specific jig construct for aligning a tibial diaphyseal fracture [8]. Additionally, the clinical application of a 3D-printed patient-specific reduction guide during MIPO of a comminuted mid-diaphyseal humeral fracture in a cat was also reported [7]. However, no published reports or studies have described the use of a 3D-printed custom reduction guide (3DRG) to align radial diaphyseal fractures using the MIPO method.

The objective of this study was to create a 3DRG for the MIPO of radial diaphyseal fractures and to evaluate the benefits of the clinical application of these guides.

## 2. Materials and Methods

### 2.1. Cadaveric Specimens

In this study, the forelimbs (*n* = 24) of 12 skeletally mature, mixed-breed canine cadavers, weighing between 6.3 and 11.5 kg, were included. All dogs had been euthanized for reasons unrelated to this study and written consent for use of the dogs was obtained from the owners. The study was approved by the institutional animal care and use committee of Jeonbuk National University (Number: JBNU 2022–0071). All cadavers were stored at −20 °C. To thaw the cadavers for the experiment, they were stored at −4 °C for 72 h before surgery.

Forequarter amputation was performed before the radiographic analysis of each forelimb, and all cadaveric forelimbs were randomly categorized into three groups using randomization software (http://www.random.org/). The forelimbs with 3DRG for the MIPO were classified as group A (*n* = 8), those with an open reduction technique were classified as group B (*n* = 8) and those with a closed reduction technique with circular external skeletal fixator (ESF) were classified as group C (*n* = 8).

A diaphyseal short oblique fracture was created in each radius at 1/3 of the radial length from the distal radial articular surface using a battery-powered oscillating saw (Colibri II, Synthes, United States) and a 6 mm saw blade. A 3 cm lateral incision was made to approach the distal radius and ulna. Metzenbaum scissors and periosteal elevators were used to expose the radius and ulna. A short oblique fracture was created in the distal radius and ulna at an angle of approximately 45 degrees in a proximolateral to distomedial direction. Subcutaneous suturing was performed routinely to close the lateral incision.

### 2.2. Data Collection

Craniocaudal and mediolateral radiographs for all forelimbs were obtained using digital radiography with a radiographic beam centered over the radial mid-diaphysis. A 1 cm diameter spherical marker, placed adjacent to the radius, was included in all images to allow image calibration. The images were viewed on a picture archiving and communication system software program (Infinitt 3.0.11.3, Infinitt Healthcare Co., Ltd., Seoul, Republic of Korea) and exported to a veterinary preoperative orthopedic planning software (vPOP-pro 2.9.2, VetSOS Education Ltd., Wales, UK) for measurement.

Radiographic measurements were performed as previously described [2,9]. Through sagittal images, the length of the radius was measured as the distance between the caudal edge of the radial head and the caudal edge of the distal radial articular surface. The proximal and distal radial anatomic axes were established in the sagittal plane and the procurvatum or recurvatum was measured. The proximal and distal radial anatomic axes were determined by a line connecting the midpoints at 10%, 20% and 30% of the radial length from the proximal and distal radial articular surfaces. Sagittal angulation (SA) was determined by the measured angle of intersection between the two axes. The varus or valgus angulation was measured in the frontal plane. The same method was used to establish the proximal and distal radial anatomic axes in the frontal plane to measure frontal angulation (FA). For the same purpose, the proximal joint reference line (PJRL) and distal joint reference line (DJRL) were established as previously described in the mediolateral and craniocaudal views [10]. In the craniocaudal view, frontal joint reference line angulation (fJLRA) was determined by the measured angle of intersection between the PJRL and DJRL. In the mediolateral view, sagittal joint reference line angulation (sJLRA) was determined in the same way. The gap width and cortical translation in the frontal (mediolateral translation) and in the sagittal (craniocaudal translation) planes were measured on the postoperative images. Fracture gap width was measured as the mean distance between the medial and lateral cortices at the fracture ends. Bone translation was measured in the same manner.

### 2.3. Production of 3DRGs

Computed tomography (CT) images of the forelimb of the dogs (Group A) were acquired (0.5 mm slice thickness; Toshiba Alexion 16; Toshiba Medical System). The resultant Digital Imaging and Communications in Medicine (DICOM) images were imported into medical image software (3D Slicer 5.6.1, National Alliance for Medical Image Computing, Boston, MA, USA). The forelimbs were individually segmented and 3D models of the bones were created. The wrap solidifies effect (region: outer surface, carve holes: enabled, minimum hole size: 10 mm) and smoothing (median smoothing method, kernel size: 10 mm) were applied to the 3D mesh. Stereolithography files of the 3D models of radius and ulna were exported to computer-aided design software (Fusion 360 16.4.0.2083, Autodesk, San Rafael, CA, USA). A virtual 3DRG was designed for each forelimb.

A proximal and distal base guide was manufactured approximately 1 cm from each articular surface; the length of each guide was 1.3 cm, and the width was 2 mm larger than the radial width on either side so that the guides could be applied through the standard craniolateral MIPO surgical portals. The central axis of the drilling and screw placement was designed as two parallel axes for each base guide in the lateral part and one parallel axis to each base guide in the medial part, at 55 degrees and 70 degrees from the frontal plane, respectively (Figure 1A). The plate grooves were designed in the base guides so that the pre-contoured plate could be precisely applied in the planned position (Figure 1B). The length of the cylinder on which the reduction guide was to be fitted was manufactured to be approximately 1.5 cm. The cylinders were designed with an inner diameter of 3.4 mm to allow for space for the screwdriver to enter and a thickness of 1.5 mm for stability. To tightly compress and fix the base guide on the bone surface, we used the lag screw technique using 1.2 mm cortical screws inside the cylinder. After placing three screws of the appropriate size, we added a hemispherical structure in which the screw head was seated and a hole structure with an inner diameter of 1.6 mm that functioned as a gliding hole (Figure 1C,D).

Subsequently, a reduction guide was designed. This was analogous to an ESF, with cylinders used to accept the cylinders on the base guide and a connecting bar. The inner diameter of the reduction guide cylinders was designed to be 0.5 mm larger than the outer diameter of the base guide cylinders so that they could be fitted onto the base guide. The reduction guides were designed for the lateral and medial side (Figure 2).

Radius models were 3D printed using a resin 3D printer (A1, Zerone, Seoul, Republic of Korea), using a dental surgical guide resin (SG-100, Graphy, Seoul, Republic of Korea). The surgical guides were 3D printed using a resin 3D printer (Pixel One, Zerone, Gyeonggi, Republic of Korea) and dental surgical guide resin. After printing, the guides were washed, dried for 30 min and cured with UV light at a wavelength of 405 nm for 60 min (3DP-100S, CUBICON, Gyeonggi, Republic of Korea). Before the cadaveric experiment, bio-rehearsal was performed using a 3D-printed bone model (Figure 3).

### 2.4. Surgical Procedure

For closed reduction with 3DRG, a craniolateral MIPO approach to the radius, as previously described, was performed at the planned site [11,12,13] (Figure 4A,B). The muscles and soft tissues at the site of placement of the base guides were dissected from the bones. The base guide was held in place manually or using bone-holding forceps. A 1.2 drill sleeve (ABLE Inc., Jeonbuk, Republic of Korea) was placed onto the base guide cylinder, and the radius was drilled by using a 0.8 drill bit (ABLE Inc.). Then, a stainless steel self-tapping 1.2 mm cortical screw (ABLE Inc.) was placed bi-cortically. As the screw head was seated on the hemispherical structure, the guides and radius were compressed, like that of the lag screw technique. After the base guides were tightly compressed and fixed on the bone surface, a lateral reduction guide was fitted on either the proximal or distal side of the base guide cylinder, followed by fracture distraction (Figure 4C). When fracture distraction was achieved, the lateral reduction guide was fitted on the base guide cylinders of the proximal fragment. Next, the medial reduction guide was fitted on the base guide cylinders on the medial side. The pre-contoured plate was placed and the locking screws were inserted (Figure 4D,E). After the plate placement, reduction was assessed using fluoroscopy (Figure 5). The open reduction technique and closed reduction technique with a circular external skeletal fixator (ESF) were performed as previously described [5,12,13,14].

### 2.5. Statistical Analysis

The analyses were performed using SPSS (version 26.0; IBM, Armonk, NY, USA). All data are reported as the mean ± SD, and differences between the three groups were compared using the Kruskal–Wallis test. The Mann–Whitney test was used for post hoc tests. Values of *p* < 0.05 were considered significant.

## 3. Results

### 3.1. Differences in Sagittal Angulation before and after Surgery of Each Radius

The mean ± SD of the difference of sagittal angulation was 2.50 ± 1.24, 1.55 ± 1.10 and 1.86 ± 1.37 for groups A, B and C, respectively. There were no significant differences among the three groups (Table 1).

### 3.2. Differences in Sagittal Joint Reference Line Angulation before and after Surgery of Each Radius

The mean ± SD of the difference in sagittal joint reference line angulation was 2.85 ± 0.72, 2.28 ± 0.67 and 2.73 ± 0.87 for groups A, B and C, respectively (Table 1). There were no significant differences among the three groups (Table 1).

### 3.3. Fracture Gap of Each Radius

The mean ± SD of the fracture gap for each group after surgery was 0.63 ± 0.20, 0.55 ± 0.19 and 0.75 ± 0.32 for groups A, B and C, respectively. There were no significant differences among the three groups (Table 1).

### 3.4. Craniocaudal Translation of Each Radius

The mean ± SD of the craniocaudal translation for each group after surgery was 0.28 ± 0.21, 0.38 ± 0.18 and 0.42 ± 0.23 for groups A, B and C, respectively. There were no significant differences among the three groups (Table 1).

### 3.5. Differences in Frontal Angulation before and after Surgery of Each Radius

The mean ± SD of the difference in frontal angulation was 1.21± 1.12, 0.85 ± 0.62 and 2.35 ± 1.26 for groups A, B and C, respectively (Table 1). There was a significant difference between groups B and C (*p* < 0.05) (Table 2).

### 3.6. Differences in Frontal Joint Reference Line Angulation before and after Surgery of Each Radius

The mean ± SD of the difference in frontal joint reference line angulation was 1.20 ± 0.64, 1.11 ± 0.33 and 1.94 ± 0.58 for groups A, B and C, respectively (Table 1). There was a significant difference between groups B and C (*p* < 0.05) (Table 2).

### 3.7. Mediolateral Translation of Each Radius

The mean ± SD of the mediolateral translation for each group after surgery was 0.68 ± 0.50, 0.25 ± 0.11 and 0.80 ± 0.37 for groups A, B and C, respectively (Table 1). There was a significant difference between groups B and C (*p* < 0.05) (Table 2).

### 3.8. Surgical Time

The mean ± SD of surgical time for each group was 98.62 ± 12.99, 115.87 ± 12.73 and 117.75 ± 15.00 for groups A, B and C, respectively (Table 1). There were significant differences between groups A and B, and between groups A and C. Group A had the shortest surgical time (Table 2).

## 4. Discussion

The present study was conducted to evaluate the use of 3DRG for MIPO of short oblique radial diaphyseal fractures in cadaveric dogs. The results show that 3DRG for MIPO of short oblique radial diaphyseal fractures is reliable for accurate fracture alignment and apposition and in reducing surgical time.

Several reports in human and veterinary studies have described the use of 3D-printed custom guides [8,11,15,16,17,18,19,20,21,22,23]. However, few have reported on the use of a 3D-printed custom reduction guide for the MIPO procedure [7,24]. The guide design in previous studies used a method of applying the guides using pins. In this study, the lag screw manner was applied instead of using pins to allow for the base guide to be tightly compressed on the bone surface. Then, the varus–valgus angulation was addressed by fitting a lateral reduction guide on the base guide to correct the valgus and a medial reduction guide on the base guide to correct for varus angulation.

In this study, the mean difference in the pre and postoperative FA and fJRLA values in group A, which used the 3D-printed patient-specific MIPO guide, was 1.21 ± 1.12 and 1.21 ± 0.64. They did not exceed 5 degrees. In addition, the mean difference between the pre- and postoperative SA and sJRLA values was 2.50 ± 1.24 and 2.85 ± 0.72. They also did not exceed 5 degrees. Compared to the values reported by Fox et al., the alignment in the frontal and sagittal planes was judged to be excellent for all fractures [25,26].

An accurate reduction in simple fractures through the MIPO method is more likely to result in translational malalignment than open reduction [25,27,28]. In our study, there were three cases with a translation greater than 1 mm in group A and four such cases in group B. However, in group C, there were no cases with a translation greater than 1 mm. The translational malalignment achieved in this study was similar to a previous study comparing MIPO and open reduction in radius–ulna fractures in dogs. This translation malalignment could only be detected radiographically and did not affect functional outcomes [2,29].

In this study, the mean total surgical time in group A was shorter than that in group B and group C. The 3DRG was designed to fit exactly onto the patient’s bones and was also firmly fitted to the bone surface in a lag screw fashion. In addition, the medial and lateral reduction guides were fitted at different angles on the base guide without the process of removing the base guide, so that the reduction was maintained firmly during the surgical operation. A typical MIPO requires fluoroscopy. The disadvantages of using fluoroscopy during surgery include radiation exposure and prolonged surgical time. In this study, we were able to minimize the use of fluoroscopy by using 3DRG. Therefore, comprehensively, group A, which used a 3D guide, showed a shorter surgical time than the other groups.

Another advantage of this 3D guide is that it contains plate grooves. The plate grooves were designed in the base guides so that the pre-contoured plate could be precisely applied in the planned position. This component could prevent repositioning or distortion of the reduction due to the displacement of the plate when the plate and screw are applied.

This study had several limitations. First, since this study was a cadaveric study, alignment and reduction in the fracture segments were unaffected by muscle contraction, callus formation, or fracture hematomas. All these factors could potentially influence fracture alignment in vivo and may interfere with accurate reduction when fixation is applied. Second, for the application of guides on patients, additional anesthesia is necessary for CT imaging to produce 3D-printed patient-specific MIPO reduction guides, presenting the disadvantage of requiring extra anesthesia. Subsequently, after the CT imaging process, it takes approximately 4 h to design and produce the guides, with sterilization requiring about 40 min using autoclaving. Due to these reasons, performing surgery using 3DRG immediately after CT imaging is impractical. This may also be a limitation of this study as the guide production time was not included in the surgical time in this study. Additionally, inserted screws compressing the base guides to the radius could potentially act as stress risers in vivo. Finally, in this study, CT scans were performed before inducing a short oblique fracture of the radius, and a guide was designed using the pre-fracture bone as a template. In a human study, the left and right human radii have differences in size and orientation of less than 1 mm or 1° in all planes [30]. In a previous study, in a case with a unilateral humerus fracture, the intact contralateral bone was mirrored through computer-aided design software and used as a template [7]. The major proximal and distal fragments are oriented in all planes to match the mirrored contralateral humerus, which was used to design patient-specific 3D MIPO guides. Although we designed the guide before fracture induction, it is considered that the guides described in this study could also be designed for patients who already have a unilateral radial fracture.

## 5. Conclusions

In conclusion, the use of a patient-specific 3D reduction guide during MIPO of short oblique radial diaphyseal fracture in dogs is reliable for the alignment and apposition of fractures and reduces surgical time.

## Figures and Tables

**Figure 1 vetsci-11-00145-f001:**
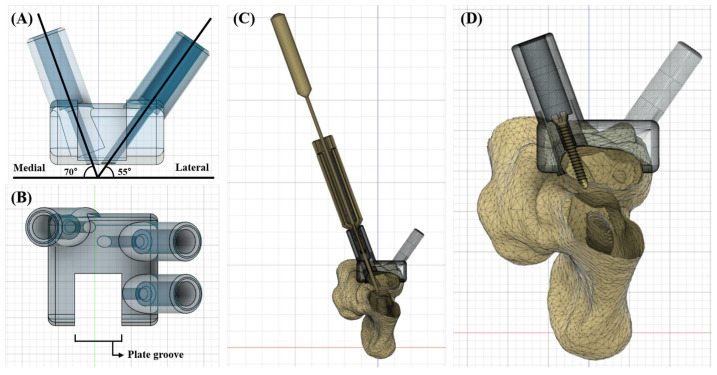
Detailed designs of a 3DRG. The central axis of the drilling and screw placement on the medial and lateral sides (**A**). The plate groove that was designed in the base guides (**B**). The base guide and radius were compressed in lag screw fashion (**C**,**D**).

**Figure 2 vetsci-11-00145-f002:**
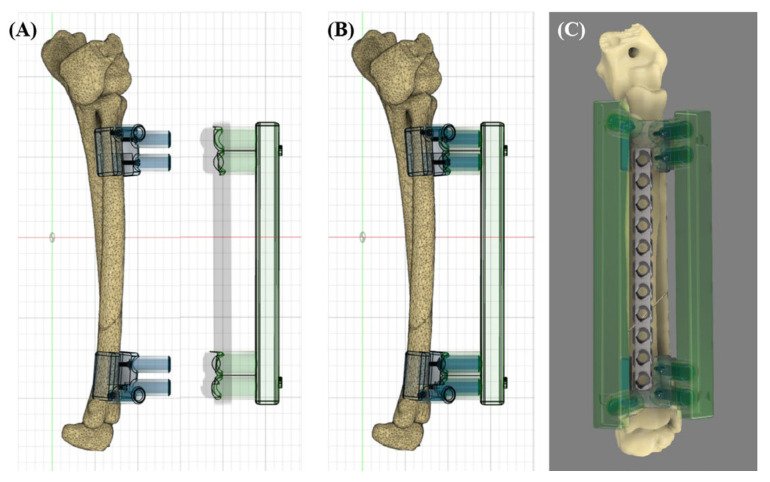
Detailed designs of the 3DRG (continued). Before fitting the reduction guide on the base guide (**A**). After fitting the reduction guide on the base guide (**B**). The 3D rendering image that 3DRG and plate were applied to (**C**).

**Figure 3 vetsci-11-00145-f003:**
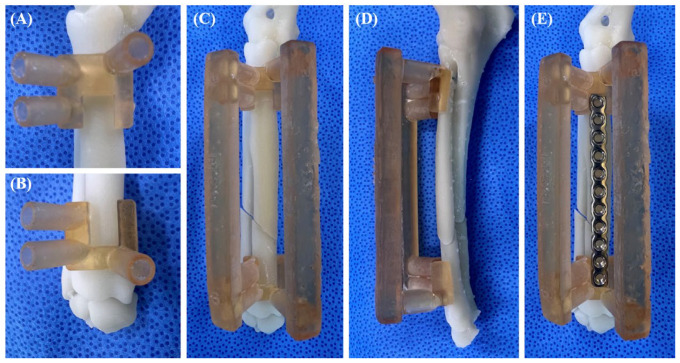
Bio-rehearsal procedure using bone model. 3DRG was placed on the 3D-printed bone for bio-rehearsal (**A**–**E**). The bone plate was pre-contoured to ensure its correct application to the plate groove (**E**).

**Figure 4 vetsci-11-00145-f004:**
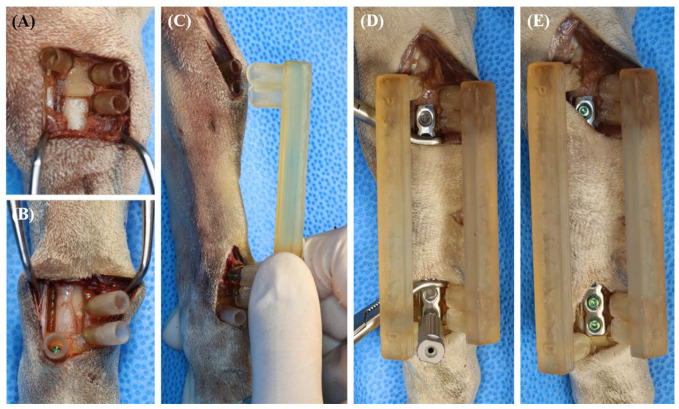
Surgical procedure. The base guides were placed on the bone of cadaveric dogs (**A**,**B**). The lateral reduction guide was first fitted on the base guide cylinders of the distal bone fragment, and then fracture distraction was performed (**C**). When fracture distraction was achieved, the lateral reduction guide was fitted on the base guide cylinders of proximal bone fragments. Next, the medial reduction guide was fitted on the base guide cylinders to ensure that the proximal and distal fracture fragments were aligned as planned (**C**,**D**). The pre-contoured plate was accurately applied in the plate groove (**E**).

**Figure 5 vetsci-11-00145-f005:**
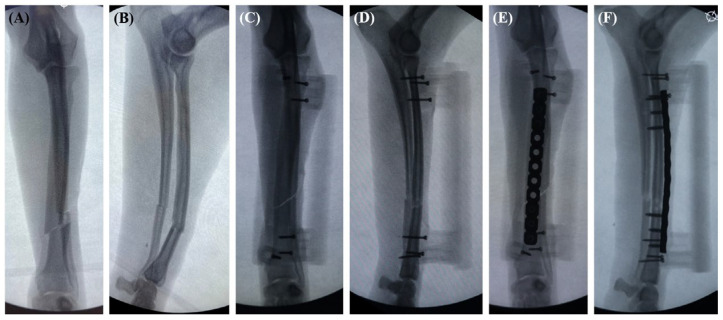
The fluoroscopic images during the procedure. Craniocaudal and mediolateral intraoperative fluoroscopic images of the antebrachium were obtained immediately after the fracture was created (**A**,**B**). Intraoperative fluoroscopic images of the antebrachium after initial application of the 3DRG (**C**,**D**). The pre-contoured plate was applied to the cranial aspect of the radius to fit the plate grooves in the base guide (**E**,**F**).

**Table 1 vetsci-11-00145-t001:** Mean (±SD) results of variables that measured the postoperative limb alignment, apposition and surgical time.

Outcome Measures	Group A	Group B	Group C	*p*-Value
Sagittal angulation (Mean difference, degree)	2.50 ± 1.24	1.55 ± 1.10	1.86 ± 1.37	0.328
Sagittal JRL angulation (Mean difference, degree)	2.85 ± 0.72	2.28 ± 0.67	2.73 ± 0.87	0.280
Craniocaudal translation (mm)	0.28 ± 0.21	0.38 ± 0.18	0.42 ± 0.23	0.387
Fracture gap (mm)	0.63 ± 0.20	0.55 ± 0.19	0.75 ± 0.32	0.487
Frontal angulation (Mean difference, degree)	1.21 ± 1.12	0.85 ± 0.62	2.35 ± 1.26	0.033
Frontal JRL angulation (Mean difference, degree)	1.20 ± 0.64	1.11 ± 0.33	1.94 ± 0.58	0.037
Mediolateral translation (mm)	0.68 ± 0.50	0.25 ± 0.11	0.80 ± 0.37	0.023
Surgical time (min)	98.62 ± 12.99	115.87 ± 12.73	117.75 ± 15.00	0.039

**Table 2 vetsci-11-00145-t002:** Results of post hoc analysis of measurements indicate significant differences between the three groups.

Outcome Measures	Group A–B (*p* Values)	Group A–C (*p* Values)	Group B–C (*p* Values)
Frontal angulation (Mean difference, degree)	0.712	0.059	0.012 *
Frontal JRL angulation (Mean difference, degree)	0.472	0.120	0.039 *
Mediolateral translation (mm)	0.111	0.673	0.030 *
Surgical time(minutes)	0.031 *	0.027 *	0.958

* *p* < 0.05.

## Data Availability

Publicly available datasets were analyzed in this study. These data can be found by contacting the corresponding author via email.

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
