# Peer review of "Evaluation of a 3D-Printed Reduction Guide for Minimally Invasive Plate Osteosynthesis of Short Oblique Radial Diaphyseal Fracture in Dogs: A Cadaveric Study"

_vetsci, 2024, doi:10.3390/vetsci11040145_

Round 1
Reviewer 1 Report
Comments and Suggestions for Authors
I have reviewed the paper titled ‘Evaluation of a 3D-printed reduction guide for minimally invasive plate osteosynthesis of short oblique radial diaphyseal fracture in dogs: a cadaveric study’. It is very interesting study for the field of veterinary orthopedic surgery. My only concern is why the study model was chosen to be in cadavers? Why didn’t they perform survival surgery and do a follow up study to the prognosis? Also Based on the iThenticate report there is a significant overlapping of texts and a more rewriting is required.
Author Response
(Reviewer 1)
I have reviewed the paper titled ‘Evaluation of a 3D-printed reduction guide for minimally invasive plate osteosynthesis of short oblique radial diaphyseal fracture in dogs: a cadaveric study’. It is very interesting study for the field of veterinary orthopedic surgery. My only concern is why the study model was chosen to be in cadavers? Why didn’t they perform survival surgery and do a follow up study to the prognosis? Also Based on the iThenticate report there is a significant overlapping of texts and a more rewriting is required.
Author's Reply to the Review Report (Reviewer 1)
The 3D guide used in this study was developed and designed to achieve a more straightforward and precise reduction for radius fractures in dogs compared to conventional methods. This study was conducted to assess its efficacy through cadaveric studies before applying it to actual clinical cases. Consideration is underway for future studies involving the application of the developed guide in living models or real clinical cases.
The authors collaboratively reviewed the manuscript's similarity index on iThenticate, and it appears that the duplication rate is high due to the frequent use of conventional expressions that overlap significantly.

Reviewer 2 Report
Comments and Suggestions for Authors
Dear Authors,
This is a very nice presentation of a modification using to the use of custom 3D guides. These reduction guides show great promise.
The reviewer has only a few comments to this paper.
Line 55 Please change to : This technique can help by allowing a faster return to function by preserving more periosteal blood supply around the fracture site and minimizing damage to soft tissues.
Lune 155 - 158 - This is a bit confusing. Is there only lag screw per device or two or three?
Line 164 - strike "the"
Table 2 is showing the p values for the different comparisons. It should say in heading Group A-B (p values) Group A-C (Pvalues) Group B-C (p values)
please delete the B<C in the and A<C and A<B in front of the p values in table 2
In Limitations please add that the inserted screws could be stress risers in a life model.
Comments on the Quality of English Language
The English seems fine and may need only minor editing.
Author Response
(Reviewer 2)
This is a very nice presentation of a modification using to the use of custom 3D guides. These reduction guides show great promise.
The reviewer has only a few comments to this paper.
Line 55 Please change to : This technique can help by allowing a faster return to function by preserving more periosteal blood supply around the fracture site and minimizing damage to soft tissues.
Lune 155 - 158 - This is a bit confusing. Is there only lag screw per device or two or three?
Line 164 - strike "the"
Table 2 is showing the p values for the different comparisons. It should say in heading Group A-B (p values) Group A-C (Pvalues) Group B-C (p values)
please delete the B<C in the and A<C and A<B in front of the p values in table 2
In Limitations please add that the inserted screws could be stress risers in a life model.
Author's Reply to the Review Report (Reviewer 2)
Line 55 Please change to : This technique can help by allowing a faster return to function by preserving more periosteal blood supply around the fracture site and minimizing damage to soft tissues.- I've modified the manuscript according to your comment (I've marked the changes in red).
Lune 155 - 158 - This is a bit confusing. Is there only lag screw per device or two or three?- Both parts of the base guide have 3 lag screws each. I've modified the manuscript according to your comment (I've marked the changes in red).
Line 164 - strike "the"- I've modified the manuscript according to your comment (I've marked the changes in red).
Table 2 is showing the p values for the different comparisons. It should say in the heading Group A-B (p values) Group A-C (Pvalues) Group B-C (p values)- I've modified the manuscript according to your comment (I've marked the changes in red).
Please delete the B<C in the and A<C and A<B in front of the p values in table 2.- I've modified the manuscript according to your comment (I've marked the changes in red).
In Limitations please add that the inserted screws could be stress risers in a life model.- In the Limitation section of the manuscript, we added the following text: "Additionally, inserted screws compressing the base guides to the radius could potentially act as stress risers in vivo." (I've marked the changes in red).

Reviewer 3 Report
Comments and Suggestions for Authors
Dear Authors,
This is a very interesting study set, it is properly presented and explained.
I have only couple of comments:
-First, it would be nice to include the data about the time needed to produce the 3D printed between the CT and the production of the instrument per se. Nothing is stated about how long it actually takes to produce this instrument and of course, sterilize it. And so discuss about it; does it requires two interventions? anesthesia for CT and 3D printing, then a second anesthesia for the surgery itself? what could happen with the patient in the meantime? or can it be done in one anesthesia? how long would it be?
-You mention that the surgery time was the only statistically significant difference with group A having the shortest, but do you include or exclude the processing of the instrument? or the potential anesthesia time/delay until it is ready? Would then be a significant difference still?
-Also, what is the cost benefit of it? does this makes the surgical technique easier? with possible less complications?
In my opinion, it would be interesting to include this as in a possible clinical setting, and application of such a method, these questions will arise.
Author Response
(Reviewer 3)
Dear Authors,
This is a very interesting study set, it is properly presented and explained.
I have only couple of comments:
-First, it would be nice to include the data about the time needed to produce the 3D printed between the CT and the production of the instrument per se. Nothing is stated about how long it actually takes to produce this instrument and of course, sterilize it. And so discuss about it; does it requires two interventions? anesthesia for CT and 3D printing, then a second anesthesia for the surgery itself? what could happen with the patient in the meantime? or can it be done in one anesthesia? how long would it be?
-You mention that the surgery time was the only statistically significant difference with group A having the shortest, but do you include or exclude the processing of the instrument? or the potential anesthesia time/delay until it is ready? Would then be a significant difference still?
-Also, what is the cost benefit of it? does this makes the surgical technique easier? with possible less complications?
In my opinion, it would be interesting to include this as in a possible clinical setting, and application of such a method, these questions will arise.
Author's Reply to the Review Report (Reviewer 3)
- On average, it takes 4 hours to make the guide and 40 minutes to sterilize it using an autoclave. Therefore, it cannot be used in surgery immediately after CT scanning, which has the disadvantage of requiring additional anesthesia for CT scanning separate from surgery. Therefore, when 3DRG is applied to real clinical cases, surgery is scheduled separately from the day of the CT scan, and the patient must be discharged or hospitalized after the CT scan.
In the manuscript, we added this to the Limitations part as follows: ‘Second, for the application of guides on patients, additional anesthesia is necessary for CT imaging to produce 3D-printed patient-specific MIPO reduction guides, presenting the disadvantage of requiring extra anesthesia. Subsequently, after the CT imaging process, it takes approximately 4 hours to design and produce the guides, with sterilization requiring about 40 minutes using autoclaving.
Due to these reasons, performing surgery using 3DRG immediately after CT imaging is impractical. This may also be a limitation of this study as the guide production time was not included in the surgical time in this study.’ (I've marked the changes in red).